# Assessing the appropriateness of the Moving Epidemic Method and WHO Average Curve Method for the syndromic surveillance of acute respiratory infection in Mauritius

Mohabeer Teeluck[1]*, Atsushi Samura[2]¤

**1** Communicable Disease Control Unit, Ministry of Health and Wellness, Port Louis, Republic of Mauritius,
**2** World Health Organization Country Office for Mauritius, Port Louis, Republic of Mauritius

☯ These authors contributed equally to this work.
¤ Current address: World Health Organization Country Office for Lao People's Democratic Republic, Vientiane, Lao People's Democratic Republic
* mopharm1981@hotmail.com

## Abstract

### Introduction

Mauritius introduced Acute respiratory infection (ARI) syndromic surveillance in 2007. The Moving Epidemic Method (MEM) and the World Health Organization Average Curve Method (WHO ACM) have been used widely in several countries to establish thresholds to determine the seasonality of acute respiratory infections. This study aimed to evaluate the appropriateness of these tools for ARI syndromic surveillance in Mauritius, where two or more waves are observed.

### Method

The proportion of attendance due to acute respiratory infections was identified as the transmissibility indicator to describe seasonality using the Moving Epidemic Method and the WHO Average Curve Method. The proportion was obtained from weekly outpatient data between 2012 and 2018 collected from the sentinel acute respiratory infections surveillance. A cross-validation analysis was carried out. The resulting indicators of the goodness of fit model were used to assess the robustness of the seasonal/epidemic threshold of both the Moving Epidemic Method and WHO Average Curve Method. Additionally, a comparative analysis examined the integrity of the thresholds without the year 2017.

### Result

The cross-validation analysis demonstrated no statistically significant differences between the means scores of the indicators when comparing the two waves/seasons curves of WHO ACM and MEM. The only exception being that the Wilcoxon sign rank test strongly supported that the specificity mean score of the two waves/seasons curve for WHO ACM outweighed that of its corresponding wave model for the MEM (P = 0.002). The comparative

**Funding:** The author(s) received no specific funding for this work.

**Competing interests:** The authors have declared that no competing interests exist.

analysis with 2017 data showed the value of the epidemic threshold remained the same regardless of the methods and the number of seasonal waves.

## Conclusion

The two waves models of the Moving Epidemic Method and WHO Average Curve Method could be deployed for acute respiratory infection syndromic surveillance in Mauritius, considering that two or more activity peaks are observed in a season.

## Introduction

Acute respiratory infection (ARI) is the self-limiting inflammation of the respiratory tract anywhere from the nose to the alveoli. It is associated with a wide range of combinations of symptoms and signs. ARI is categorised as upper respiratory tract infection and lower respiratory tract infection [1].

According to the Global Burden of Disease Study 2017, ARI is among the leading cause of mortality, with global figures estimating about 2.6 million fatalities per year [2]. Most acute respiratory infection is caused by bacterial agents, respiratory viruses such as adenoviruses, influenza A and B viruses, parainfluenza viruses, respiratory syncytial viruses, coronaviruses, human enteroviruses and human rhinoviruses [3].

Mauritius introduced a syndromic surveillance system in 2007 with nine sentinel sites to monitor ARI, gastroenteritis, and conjunctivitis, as these are often accountable for public health emergencies [4]. ARI attendances and admissions in regional hospitals have been used as an indicator to assess the seasonality of influenza in Mauritius since 2011. The utilisation of ARI as a proxy for influenza surveillance is common in countries where influenza-like illness (ILI) data is limited or unavailable [5].

In temperate countries, seasonal influenza peaks are observed in the winter months (November-February) in the northern hemisphere and (May-October) in the southern hemisphere. In tropical and sub-tropical countries, including Mauritius, there tend to be multiple peaks of activity throughout the year, associated with the rainy season in summer and the dry winter [6–8]. Typically, in temperate countries for the northern hemisphere, the season happens between week 40 and week 20 and vice versa (week 20 to week 40) for southern hemisphere countries [9].

After the 2009 H1N1 pandemic, a report of the Review Committee on the Functioning of the International Health Regulations (2005) and on Pandemic (H1N1) 2009 identified a major challenge in the timely assessment of global influenza severity due to the lack of robust surveillance systems and the limited availability of comparable standardised data [10]. Consequently, WHO elaborated the Pandemic Influenza Severity Assessment (PISA), a framework which defines "the severity of influenza using three indicators: (i) transmissibility, (ii) seriousness of the disease and (iii) impact" [11]. The standardised case definitions of ARI, ILI, severe acute respiratory infections (SARI) are typically used to assess a particular influenza season [12–15].

In Mauritius, the Ministry of Health and Wellness (MOHW) created a summary seasonality curve averaging the proportion of attendance for ARI since 2013, for monitoring of ARI activity and as a proxy for influenza seasonality. However, this may not describe what a hypothetical season will look like. WHO and European Centre for Disease Prevention and Control recommended two tools to strengthen influenza surveillance: the Moving Epidemic Method (MEM) and the WHO Average Curve Method. MEM has been developed by the Health

Sentinel Network of Castilla y Leon (Spain) and available as a web application or as an R package [14,16]. The Global Epidemiological Surveillance Standards for Influenza manual was published in 2013 and established international standards to support member states with collecting, reporting, and analysing data from inpatient and outpatient respiratory disease surveillance. The manual also proposed the Average Curve Method adopted by several countries to set up thresholds [17–19].

Hence, this study aims to evaluate the appropriateness of MEM and the WHO Average Curve Method in Mauritius, where more than one peaks per season are observed, using epidemiological ARI data as a measure to proximate influenza transmissibility.

## Methods

### Ethical statement

This study does not need ethical approval with regards to public health surveillance activities. All data are already anonymous at data sharing process within the sentinel surveillance system.

### Epidemiological data

ARI has been defined as a case presenting with laryngitis, pharyngitis, rhinitis, tonsillitis, bronchitis, common cold, influenza or pneumonia and may or may not be associated with one or more of the following symptoms: fever, cough, running nose and difficulty breathing.

ARI data are collected from nine sentinel sites (five regional hospitals, two district hospitals, one community hospitals and the Ear-Nose-Throat centre), which are distributed across all the subnational regions in Mauritius Island (hereunder referred as Mauritius) [20]. The following ARI data between 2012 and 2018 were collected at each sentinel site and reported to the Central Records Division, MOHW weekly: 1) Total number of outpatient visits (attendance at the Accident and Emergency department and Outpatient department) and 2) Number of attendance for cases of ARI. The proportion of attendance for ARI (PropARI) for each week between 2012 and 2018 was chosen as an indicator for transmissibility [19]. The numerator for PropARI is the number of attendance for ARI, and the denominator is the total number of outpatient visits from the Accident and Emergency department and Outpatient department.

Since the number of processed virological specimens was very limited in the past eight years, virological data and its deliverable (e.g. influenza positivity rate) were not used in this study.

### Moving Epidemic Method (MEM)

The main characteristic of MEM is dividing the season into a pre-epidemic, epidemic and post-epidemic period. There is a web application of MEM, which we used for this study [21,22] (Version 2.15). The web app also calculates the seasonal/epidemic threshold and other alerts/intensity levels based on uploaded historical data.

An excel spreadsheet containing the weekly rates of PropARI was calculated between 2012 and 2018 and were uploaded on the web app. Since multiple waves were observed every year in the averaging curve of the proportion of ARI in outpatient services, the two-waves (observed) transformation method was chosen for the threshold calculation. The preferred parameters to calculate the epidemic threshold was the median and Nyblom confidence interval, and the intensity threshold was created using the arithmetic mean and point confidence interval. We chose the median and Nyblom CI for the seasonal threshold as we felt it would be more appropriate to compare MEM with the WHO Average Curve method (which calculated

the epidemic thresholds using the median formula on an Excel spreadsheet). Additionally, when the sample size is large enough, the MEM technical manual recommends the use of the Hettmansperger and Sheather (1986) and Nyblom (1992) method [23]. Vega et al. stated that the choice of parameters for establishing thresholds using the MEM method will not alter "the underlying structure of the model" and is customisable to various settings [14]. Therefore, we had the flexibility to choose this parameter based on the fact that the sample size was greater than 5. The goodness indicators for MEM (Sensitivity, Specificity, Positive Predictive Value and Negative Predictive Value) in this study were optimised using a slope parameter of 2.2 [23].

## WHO average curve method (WHO ACM)

WHO average curve method (WHO ACM) can be done on MS Excel, following the WHO Global Epidemiological Surveillance Standards for Influenza manual [19]. However, The WHO Average Curve Shiny App [24,25] (Test version: 0.3) provided thorough modelling options where users can choose a model for one or two waves per season according to their data. The same dataset of the PropARI rates (2012 to 2018) was uploaded on the Shiny App to calculate the thresholds. We used the median of the PropARI (2012 to 2018) as the epidemic threshold which defines the beginning and end of the season. The processed data for PropARI was then extracted on MS Excel to formulate the two waves curve to compare with the corresponding two waves transformation for MEM.

## Establishing thresholds

MOHW held a stakeholder meeting in August 2018 where both the WHO ACM and MEM were recognised as appropriate methods to examine ARI activity in Mauritius. In terms of the seasonality assessment, two rules are commonly used to announce the onset of the influenza season [18]: 1) The first-week-declaration rule where onset is announced once threshold is crossed on the first week; 2) The two consecutive-week-declaration rule where onset is declared once the threshold is crossed for two consecutive weeks. The second option was chosen as this was more conservative in identifying the seasonality. The metrics to define the following thresholds were set as the upper limits of one-sided confidence intervals from the normal distribution [18,19,26]:

- The upper limit of the 40% confidence interval was designated as the moderate intensity threshold

- The upper limit of the 90% confidence interval was labelled as the high intensity threshold

- The upper limit of the 97.5% confidence interval was chosen as the very high intensity threshold.

    Lastly, ARI activity levels were also defined below:

- No activity: Any activity below the seasonal/epidemic threshold

- In season: Any activity between the seasonal/epidemic and moderate alert level

- Moderate: Any activity between the moderate and high alert level

- High: Any activity between the high and very high alert level

- Extraordinary: Any activity beyond the very high alert level

### Comparative analysis without the year 2017

A comparative analysis was applied to the two waves model of both MEM and WHO ACM by excluding the year 2017. This process was undertaken to examine the integrity of the thresholds, as the proportion of attendance for ARI in 2017 was exceptionally high.

### Cross-validation analysis

The cross-validation technique was used to assess the robustness of the two waves/seasons curves of PropARI (2012 to 2018) with MEM and WHO ACM [14]. To determine the goodness of fit of both waves model, the following indicators were used [14,27]

1. Sensitivity: The number of epidemic weeks above the epidemic threshold divided by the number of epidemic weeks (epidemic length).

2. Specificity: The number of non-epidemic weeks below the epidemic threshold divided by the number of non-epidemic weeks.

3. Positive predictive value (PPV): The number of epidemic weeks above the epidemic threshold divided by the number of weeks above the threshold.

4. Negative predictive value (NPV): The number of non-epidemic weeks below the epidemic threshold divided by the number of weeks below the threshold.

5. Timeliness: The number of weeks between the alert week (the first week above seasonal/epidemic threshold) and the first week of the epidemic period as modelled by MEM and WHO ACM.

### Statistical analysis

The Wilcoxon signed-rank test was designated to differentiate between the mean scores of the cross-validation indicators except for timeliness, where its median difference was estimated. The choice of the Wilcoxon signed-rank test was due to its robustness when assuming that normality does not exist. Stata IC Version 13 was used to conduct these statistical tests [28]. The analysis was carried out using the following Stata command, signed-rank mean score 1 = mean score 2. The thresholds for significance (alpha) was 0.05.

### Comparison of ARI activity levels between MEM and WHO ACM

To compare the implication of MEM and WHO ACM in the past data, we created a calendar demonstrating the ARI activity levels in each epidemiological week from 2012 to 2018, which provided narrative analysis beyond the quantitative comparison between the two modelling methods.

## Results

### Characteristics of the thresholds and epidemic period for the two wave/season curve of MEM and WHO ACM

The thresholds of PropARI for the two wave/season (W/S) curve for MEM varied between 0.103 (epidemic threshold) and 0.241 (very high threshold). The associating thresholds for the average 2WS curve of WHO ACM ranged between 0.134 (epidemic threshold) and 0.223 (very high threshold). A variation of 23 per cent was observed between the epidemic thresholds of the 2WS curves of MEM and WHO ACM. The medium thresholds for the 2WS curve of MEM and WHO ACM were equal in value. The differences between the high and very high

**Table 1. Characteristics of the thresholds for MEM (with and without 2017) and WHO ACM (with and without 2017), by the different types of waves detection methods, Mauritius, ARI season 2012–2018.**

| Epidemic detection method | Waves detection model | Epidemic threshold | Moderate threshold | High threshold | Very High threshold | Peak value | Peak week | Epidemic Start week | Epidemic End week | Epidemic length |
|---|---|---|---|---|---|---|---|---|---|---|
| MEM[a] | 2WS curve[e] | 0.103 | 0.170 | 0.219 | 0.241 | 0.156 | 25 | 15 | 38 | 23 |
| MEM[a] w/o 2017[b] | 2WS curve[e] | 0.103 | 0.165 | 0.194 | 0.207 | 0.150 | 29 | 14 | 37 | 23 |
| WHO ACM[c] | 2WS curve[e] Wave 1 | 0.134 | 0.176 | 0.231 | 0.255 | 0.185 | 24 | 19 | 43 | 24 |
| | 2WS curve[e] Wave 2 | 0.134 | 0.164 | 0.183 | 0.191 | | | | | |
| | Average 2WS curve[e] | 0.134 | 0.170 | 0.207 | 0.223 | | | | | |
| WHO ACM w/o 2017[d] | 2WS curve[e] Wave 1 | 0.134 | 0.169 | 0.200 | 0.214 | 0.174 | 24 | 18 | 43 | 25 |
| | 2WS curve[e] Wave 2 | 0.134 | 0.160 | 0.173 | 0.179 | | | | | |
| | Average 2WS curve[e] | 0.134 | 0.165 | 0.187 | 0.197 | | | | | |

[a]Moving Epidemic Method.

[b]Moving Epidemic Method without the year 2017.

[c]WHO Average Curve Method.

[d]WHO Average Curve Method without the year 2017.

[e]Two waves/seasons curve.

thresholds for MEM and WHO ACM were five and seven per cent, respectively. The epidemic period for the 2WS curve for MEM (23 weeks) was a week shorter than the associating curve (24 weeks) of WHO ACM. These characteristics are summarised in Table 1.

## Comparative analysis for MEM without the year 2017 (MEM w/o2017)

The thresholds for MEM w/o 2017 extended from 0.103 (epidemic threshold) to 0.207 (very high threshold) for the 2WS curve (Table 1).

The value of the epidemic threshold from MEM (**Fig 1**A) to MEM w/o 2017 (**Fig 1**B) displayed no change for the 2WS curve. The gap between the intensity thresholds for the 2WS curve dropped by seven per cent (in-season) and 41 per cent for both moderate and high ARI activity zones when 2017 was excluded (**Fig 1**).

## Comparative analysis for WHO ACM without the year 2017 (WHO ACM w/o 2017)

The thresholds for the 2WS curve of WHO ACM w/o 2017 spanned from 0.134 (epidemic threshold) to 0.197 (very high threshold) (Table 1).

The epidemic baselines (0.134) of the 2WS curve for WHO ACM w/o 2017 (Fig 2B) were equivalent to the epidemic thresholds of its corresponding curve for WHO ACM (Fig 2A). The area between the intensity thresholds for the 2WS curve narrowed by 14 per cent (in-season), 41 per cent (moderate) and 38 per cent (high) without the year 2017 (Fig 2)

Fig 3 depicted the time-series for the two wave/season curve of the proportion of attendance for ARI using MEM for the season 2012 to 2018. From 2012 to 2014 and 2016 to 2017, Prop-pARI crossed over the 40% confidence interval, with peak ARI activities in the moderate zone. An unusual ARI event was recorded in 2017 when the PropARI rate was optimal at week 23

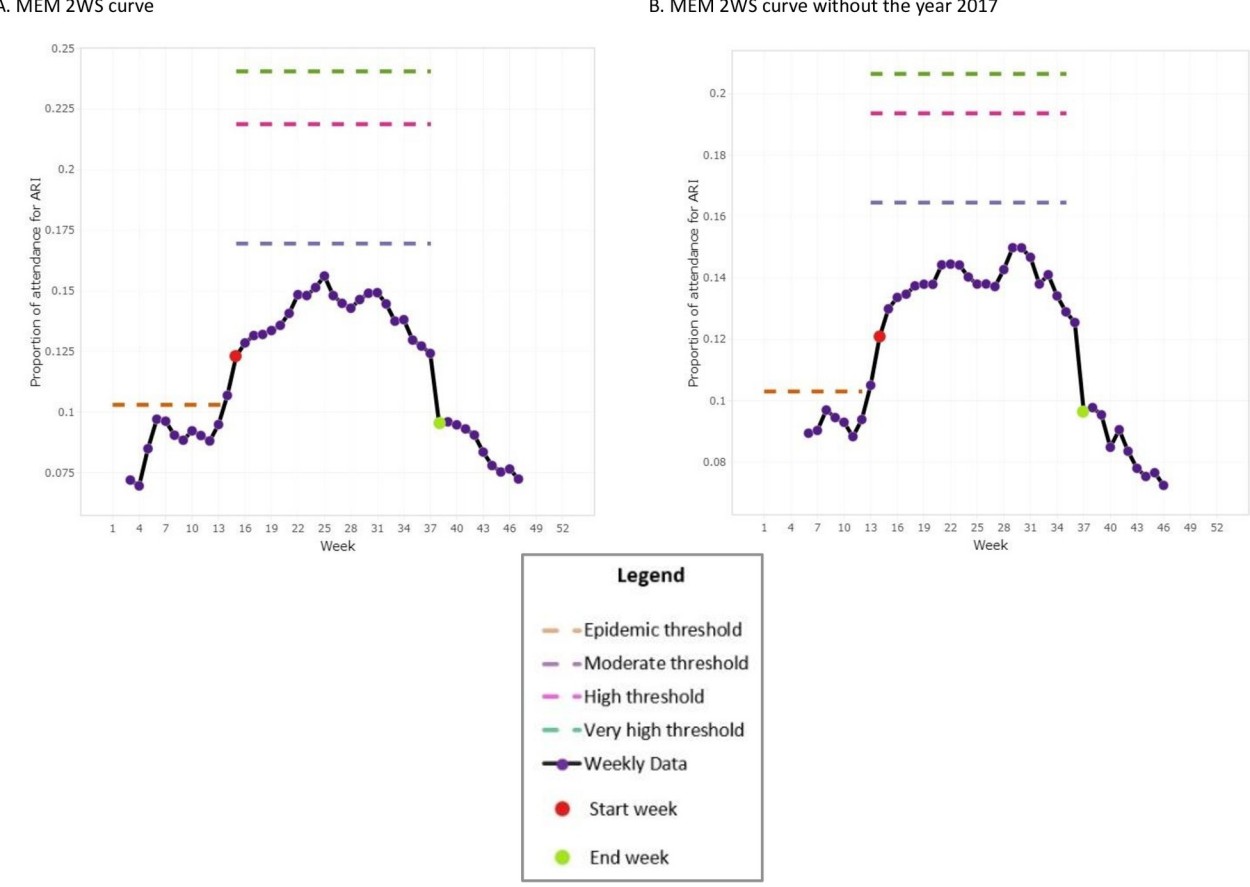

**Fig 1. Comparative analysis for the two wave/season curve of the proportion of attendance for ARI in Mauritius using MEM.**

beyond the very high-level threshold. Two additional peaks were also noted in the moderate region for the 2WS curve of 2017. Except for 2017, ARI activity was highest in 2012 and was consecutively followed by 2014, 2016 and 2013. All peaks for PropARI were located within the in-season zone in 2015 and 2018.

Fig 4 portrayed the time-series for the two wave curves of the proportion of attendance for ARI from 2012 to 2018, using the WHO ACM. The two-wave method for the WHO ACM was generated as a typical season in Mauritius consisted of multiple surges in ARI activity. The intensity thresholds for the first and second wave were established to identify the peaks in their associated zone of activity, as shown in Fig 4. The first wave for the year 2012, 2013, 2014, 2016 and 2017 demonstrated peak activity in the moderate zone. In 2017, PropARI was at its upper-most in the extraordinary zone and followed by the higher intensity in 2012, 2014, 2013 and 2016. ARI activity was within the seasonal region for 2015 and 2018. The second part of the season featured additional surges in activity in the moderate (for 2012, 2013, 2015, 2016 and 2017) and upper zones (2017) for the second wave.

The 2WS curve with WHO ACM observed the highest number of the peaks in 2013 (four), followed by three in 2017 and two in 2012, 2014 and 2016. The year 2015 exhibited only one peak in the moderate zone, and no activity was recorded beyond the moderate threshold for 2018 (Fig 4). The second set of thresholds revealed additional peaks for 2013 (three), one in the moderate zone for 2015 and one in the high activity region for 2017.

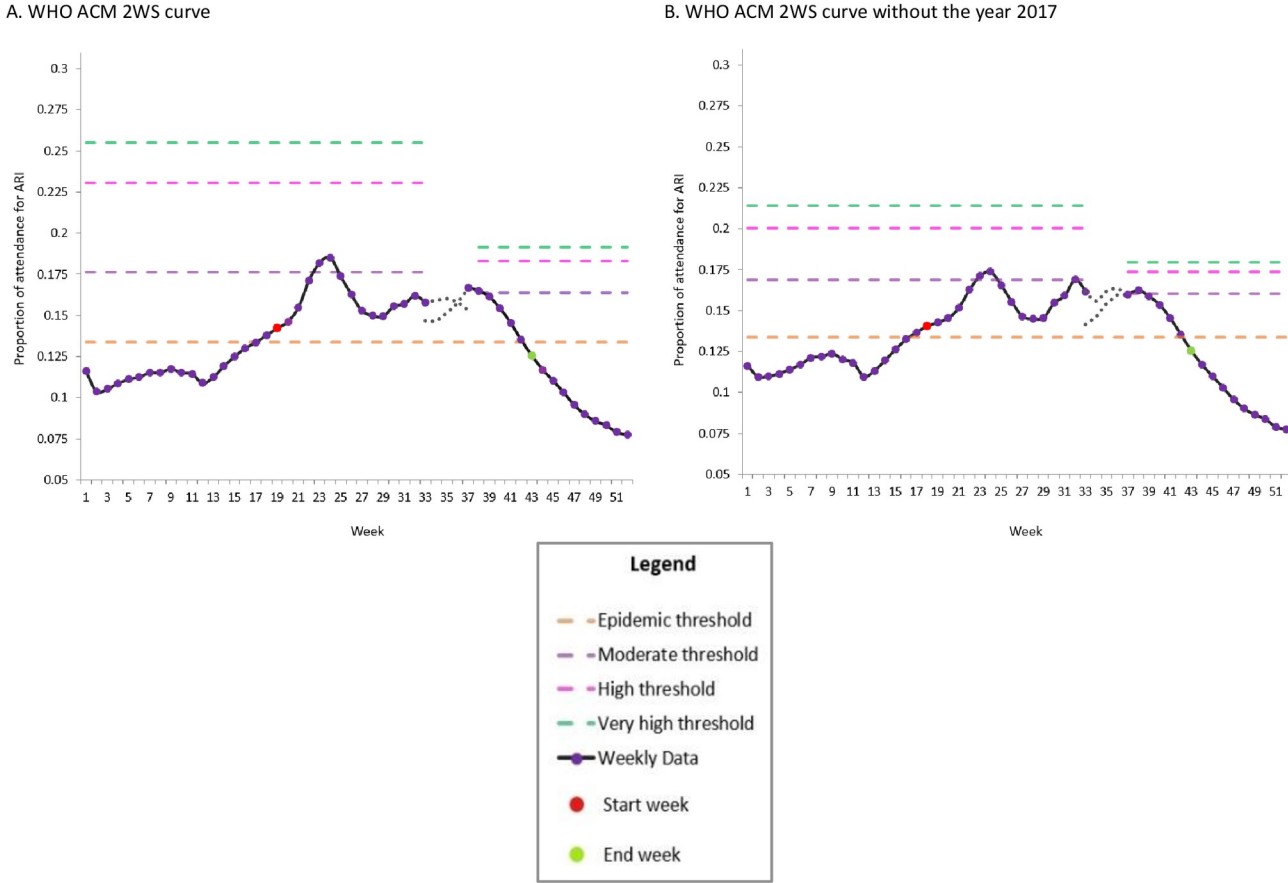

**Fig 2. Comparative analysis for the two wave/season curve of the proportion of attendance for ARI in Mauritius using WHO ACM.**

## Cross-validation analysis

We compared the 2WS curve for MEM against the 2WS curve for WHO ACM as the cross-validation analysis since no appropriate standard analytical method was challenged in this

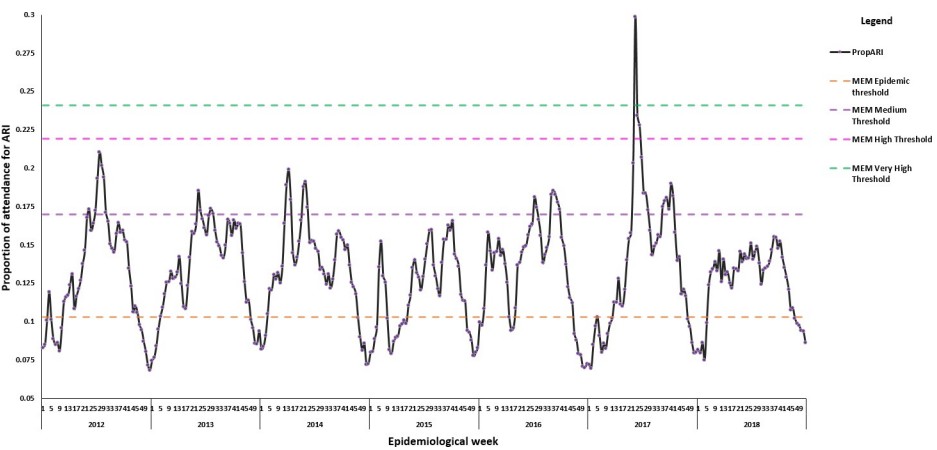

**Fig 3. Time-series for the two wave/season curve of the proportion of attendance for ARI using MEM, Mauritius, season 2012–2018.**

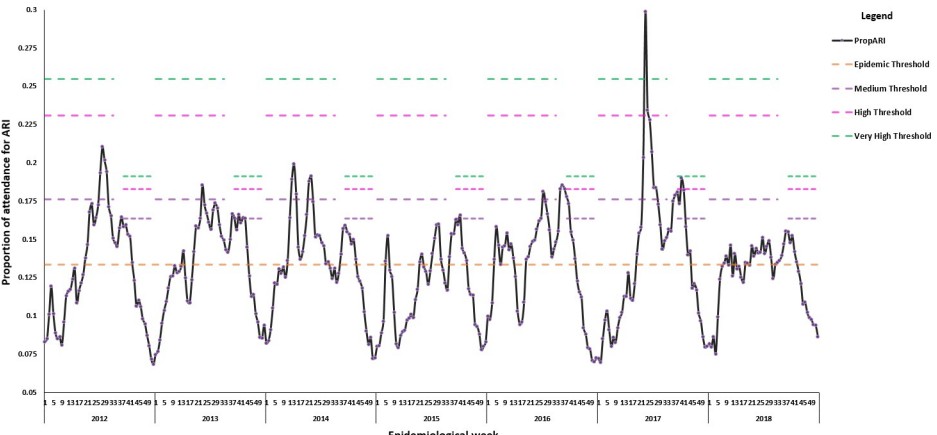

**Fig 4. Time series for the two wave curve of the proportion of attendance for ARI using WHO Average Curve Method, Mauritius, season 2012–2018.**

study's context (Table 2). The sensitivity of the 2WS curve for MEM was 0.112 units above that of the 2WS curve for WHO ACM. The p-value of 0.050 implied weak evidence to support the difference between the sensitivity mean scores of the two waves methods. At the point estimates, the specificity of the 2WS curve for MEM was 0.036 units below that of the 2WS curve for WHO ACM, which indicates strong evidence of a difference between the two models (p-value = 0.018). This strongly supported that the specificity mean score of the 2WS curve for WHO ACM outweighed its corresponding wave model for MEM. There is very weak evidence of a difference in the positive predictive values between the 2WS curve for MEM and WHO ACM (p-value = 0.063). A p-value of 0.050 demonstrates that the statistically significant differences in negative predictive values (NPV) between the 2WS curve for MEM and for WHO ACM is negligible. The median timeliness of the 2WS curve for MEM was equivalent to the 2WS curve for WHO ACM (2 weeks). Lastly, there was little evidence of differences in the median timeliness score between both 2WS curves (p-value for the Wilcoxon signed-rank test = 0.9136).

**Table 2. Comparing the indicators of the goodness of fit model for the two wave/season curves detection methods between MEM and WHO Average Curve Method, Mauritius, ARI season 2012–2018.**

| Epidemic detection methods | Waves detection model | Sensitivity | Specificity | PPV[f] | NPV[g] | Median Timeliness |
|---|---|---|---|---|---|---|
| MEM[a] | 2WS curve[e] | 0.955 | 0.908 | 0.970 | 0.893 | 2 |
| MEM w/o 2017[b] | 2WS curve[e] | 0.947 | 0.901 | 0.970 | 0.875 | 2 |
| WHO ACM[c] | 2WS curve[e] (p-value)[¶] | 0.843 (0.050) | 0.944 (0.018) | 0.955 (0.063) | 0.791 (0.050) | 2 (0.9136) |
| WHO ACM w/o 2017[d] | 2WS curve[e] | 0.816 | 0.941 | 0.954 | 0.756 | 2 |

[a]Moving Epidemic Method.

[b]Moving Epidemic Method without the year 2017.

[c]WHO Average Curve Method.

[d]WHO Average Curve Method without the year 2017.

[e]Two waves/seasons curve.

[f]Positive predictive value.

[g]Negative predictive value.

[¶]P-values for the indicators of the goodness of fit model for the two wave/season curves detection methods between MEM and WHO Average Curve Method, Mauritius, ARI season 2012–2018. Calculated with Wilcoxon sign rank test.

## Comparative analysis

Within each modelling method, MEM and WHO ACM, we conducted a comparative analysis to compare the results derived from the data between 2012 and 2018 to those of the same data-set without the year 2017 when an unusually high ARI activity was observed.

In comparing the 2WS curve for MEM against the 2WS curve for MEM w/o 2017, any alterations in the average scores of the cross-validation indicators mainly were negligible as the epidemic baselines of the 2WS curves were equal to 0.103. As the mean score difference between the indicators was zero, the t tests' values proved to be undefined.

In comparing the 2WS curves for WHO ACM against the 2WS curves of WHO ACM w/o 2017, the indicators' mean scores of the 2WS curves for WHO ACM showed slight deviation from their corresponding waves models for WHO ACM w/o 2017. The epidemic thresholds of the waves detection methods for both WHO ACM and WHO ACM w/o 2017 were equal to 0.134. The fact that the mean score difference between the cross-validation indicators were zero implied that the calculation of the results of the t-test would be indeterminate. Therefore, it was statistically not feasible to verify any significant differences between the 2WS curves for WHO ACM and WHO ACM w/o 2017.

## Comparison between the two waves detection methods for MEM and WHO ACM

Fig 5 characterised the epidemic length of ARI activities identified by two sets of thresholds based on MEM and WHO ACM over the epidemiological calendar from 2012 to 2018. The

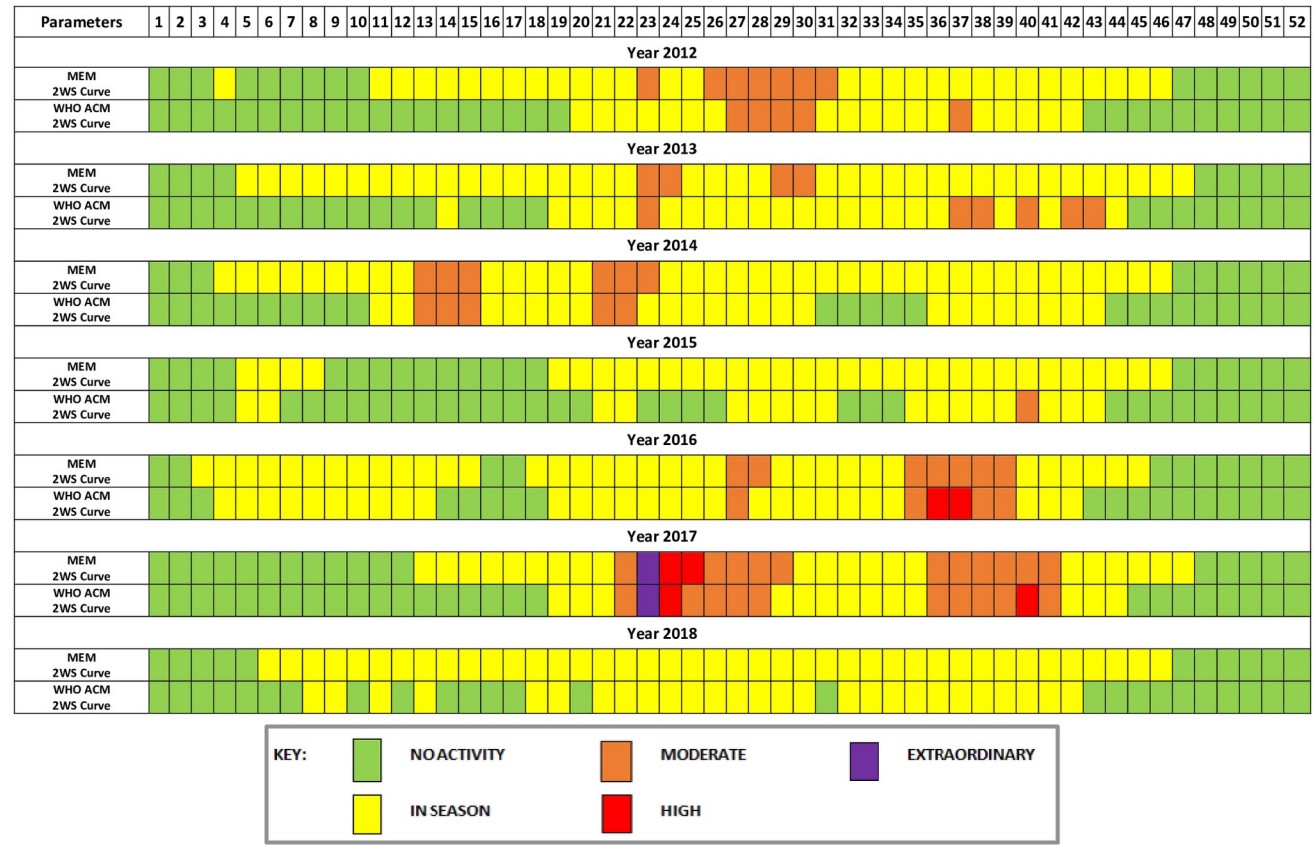

**Fig 5. The weekly proportion of attendance for ARI by parameter based on intensity zone of ARI activity of World Health Organization (WHO) method and moving epidemic method (MEM), Mauritius, seasons 2012–2018.**

most obvious observation from Fig 5 was a particularly extended epidemic period of the 2WS curve of MEM compared to the 2WS model of WHO ACM.

## Discussion

This study explored appropriate models and thresholds to monitor ARI activities in Mauritius by MEM and WHO ACM. The seasonality of Mauritius's ARI and climate was taken into consideration as factors that led to choosing the two W/S templates for this study.

In general, sensitive models allow timely detection of an outbreak and assess the magnitude of an epidemic. The specificity is also crucial in minimising the number of false positives, thus providing a true estimation of the burden of the disease [29–31]. Vega et al. and Rakocevic et al. justified using standardised epidemiological indicators instead of virological data in MEM in case of the limited virological laboratory capacity [14,27]. On the other hand, numerous studies have recommended incorporating laboratory data with epidemiological parameters as it would amplify the robustness of the thresholds [17,18]. AbdElGawad et al. examined the effect of the composite of "screened ILI consultation rate × influenza-positive percentage among sampled ILI patients" on the intensity thresholds. It was a highly sensitive and specific parameter in demonstrating the true seasonality of influenza in Egypt [26]. It is vital to investigate the viability of the composite of virological positivity rate of influenza with the proportion of attendance for ARI and ILI once these data become adequately available in Mauritius.

### ARI as a proxy for influenza severity assessment

The choice of the proportion of attendance for ARI as the measure for transmissibility was due to the limited availability of data on ILI, SARI and virological tests. The significant advantages of the ARI syndromic surveillance are its high sensitivity since laboratory confirmation is not required, flexibility and timeliness in gathering data. However, such a surveillance system generally has lower specificity, which could trigger false alerts. Vega et al. suggested that indicators such as ARI with a broader case definition will decrease the MEM model's specificity compared to ILI's "highly sensitive inclusion criteria" [30,31]. The enhanced sensitivity and specificity of the reviewed case definitions of ILI and SARI by WHO in 2011 emphasised the practicality of ILI as the ideal transmissibility indicator [29].

### Data representativeness of the ARI activity among the general population

Nine sentinel sites are geographically well distributed across the island. Although they are categorised as secondary to tertiary hospitals, they are highly accessible from both urban and rural population due to the geographical proximity and provision of free services. Therefore, collected data from these sites is considered well represent the ARI activities among the general population.

### Local climate and ARI activity

Mauritius is a tropical island that consists of two seasons: a warm, humid summer extending from November to April and a relatively cool dry winter from June to September. The warmest months are January and February, while the coldest months are July and August. The wettest months are February and March, while the driest month is October [32]. Typically, several ARI peaks have been identified throughout the year, with a more noticeable increase in activity during the winter period from May to October.

### Analytical methods and validation of the findings

According to the WHO and European Centre for Disease Prevention and Control recommendations, MEM has proven to be a practical tool for influenza surveillance. Several studies have been formulated in European countries, highlighting MEM's suitability in syndromic (ILI and ARI) and virological surveillance in temperate climates [14–16,27,33–35]. A study by Vette et al. was carried out to establish thresholds and parameters for pandemic influenza severity assessment in Australia using MEM. In this study, a one wave temperate model was preferred as most of its population reside in temperate regions, even though 40 percent of Australia's landmass consists of a tropical climate [12].

The World Health Organization endorsed the average curve method to support the setting up of intensity thresholds for influenza surveillance. Various publications from the Philippines, Cambodia, Australia (State of Victoria) and Egypt examined how to establish thresholds using the WHO proposed method [17,18,26,36]. For instance, AbdElGawad et al. evaluated the influence of influenza parameters on the values of the intensity baselines using the country-specific statistical, empirical method, WHO ACM and MEM [26].

Since Mauritius has not established its empirical thresholds and the country is affected by the tropical climate, there were no previous studies where their method is thoroughly applicable to Mauritius as a gold standard. Therefore, two different waves models based on MEM and WHO ACM were compared against each other with cross-validation.

Efficient syndromic surveillance systems should timely detect events with fewer false positives clusters. The cross-validation analysis (Table 2) suggested that the 2WS curve for WHO ACM demonstrates higher specificity that would be most relevant in detecting true events of ARI more effectively than MEM in Mauritius. In the context of achieving a balance between sensitivity and specificity for the ARI syndromic surveillance system in Mauritius, we validated the deployment of both the 2WS model for MEM and WHO ACM method (Table 2).

### Comparative analysis

The comparative analysis results (Table 1) showed that the inclusion of 2017 data caused an increase in most threshold values. However, the epidemic threshold remained the same regardless of the methods and the number of seasonal waves, while very high threshold observed the largest jump. Since a surge in 2017 was detected by the models based on both datasets, either including 2017 data or not, the models we created in this paper are robust enough to apply to the ARI data in Mauritius.

### Choice of a better fitting model in local ARI data

The extended epidemic length in MEM was explained by the lower value of the epidemic threshold in MEM than the one in WHO ACM. The lower intensity threshold values of the 2WS curve for WHO ACM heightened its ability to maximise the coverage of supplementary ARI occurrences in the different zones. The two-wave representation for WHO ACM comprised a separate set of intensity thresholds for the second part of the season (Fig 4). This distinct feature improved its specificity in recognising moderate and high ARI activities (2012, 2013, 2015–2017), which was unnoticeable in the 2WS curve for MEM (Fig 5).

### Limitations

A literature search on PubMed did not find any research papers regarding implementing two waves transformation for MEM for syndromic surveillance of respiratory diseases in a tropical country as of 8[th] July 2019. On the other hand, the proposed WHO method was integrated

within the influenza surveillance for the Philippines and Cambodia, and both countries have tropical/subtropical climates [18,36]. Despite this, we could not benchmark the findings of this research with these publications because our study focussed on the consequences of the two wave models of MEM and WHO ACM on the thresholds. This is the first limitation of this research.

Second, we could not assess the indicators of the WHO Pandemic Influenza Severity Assessment (PISA) project due to the limited availability of ILI and SARI data. This was because the national influenza sentinel surveillance protocol established the ILI and SARI case definitions in 2019 and was integrated into the existing syndromic surveillance system. However, since a study by Hall et al. has demonstrated an association between a surge in influenza activity and emergency/outpatient visits and hospitalisations for respiratory diseases, this study's findings might still be helpful to understand the influenza activities [37].

Third, although virological data has been collected since 2012, the volume of processed specimens was too small to calculate an unbiased, positive rate. The National Influenza Sentinel Surveillance protocol in Mauritius provides instruction on the virological specimen collection to strengthen the virological component of the surveillance system.

## Conclusion

Mauritius is a tropical island, and unlike temperate countries, there are several peaks of seasonal influenza that are identified during a season. Since ILI and SARI surveillance started in 2019 and virological data was inadequate, the proportion of attendance for ARI was used to proximate influenza transmissibility. Nonetheless, the parameter choice was the best among the available but suboptimal to solely reflect the influenza activity.

The two waves models seem to fit well in the local context of two or more ARI activity peaks observed in a season. The 2WS curves for MEM and WHO ACM enabled the detection of seasons and exceptional intensity. Based on our findings, we concluded that although the two wave method for WHO ACM was more specific than the 2WS model for MEM, we still recommended that both WHO ACM and MEM be deployed for ARI syndromic surveillance in Mauritius.

This study paved the way for further research to be carried out using more sensitive and specific case definitions such as ILI and appraising their application using the two W/S curve method of MEM and WHO ACM in Mauritius.

## Supporting information

**S1 Data. MEM and WHO Average Curve Method-style data.**
(CSV)

**S1 Code. Stata code for Wilcoxon signed-rank test: Signed-rank mean score 1 = mean score 2.**
(TXT)

## Acknowledgments

We are grateful to Dr Julia Fitzner for her technical advice, Global Influenza Programme, Geneva, World Health Organization, to the Department of Public Health and Central Health Records Division, Ministry of Health and Wellness and the surveillance sites for their contributed data and to Global Influenza Programme, Geneva, World Health Organization for providing the permission to use the Shiny App.

## Author Contributions

**Conceptualization:** Mohabeer Teeluck, Atsushi Samura.

**Formal analysis:** Mohabeer Teeluck.

**Investigation:** Mohabeer Teeluck, Atsushi Samura.

**Methodology:** Mohabeer Teeluck, Atsushi Samura.

**Visualization:** Mohabeer Teeluck, Atsushi Samura.

**Writing – original draft:** Mohabeer Teeluck, Atsushi Samura.

**Writing – review & editing:** Mohabeer Teeluck, Atsushi Samura.

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
