## [Decision Letter · Decision Letter 0]

4 Feb 2021

PONE-D-20-28117

Assessing the appropriateness of the Moving Epidemic Method and WHO Average Curve Method, for the syndromic surveillance of acute respiratory infection in Mauritius

PLOS ONE

Dear Dr. Teeluck,

Thank you for submitting your manuscript to PLOS ONE. After careful consideration, we feel that it has merit but does not fully meet PLOS ONE’s publication criteria as it currently stands. Therefore, we invite you to submit a revised version of the manuscript that addresses the points raised during the review process.

A minor revision is suggested, as no significative change of methods was deemed necessary. But the presentation of data and the writing needs considerable and detailed review. 

This manuscript falls into a subject area - public health surveillance - in which it has been extremely difficult to find reviewers due to the overall increased workload for all experts in the field associated with the COVID pandemic. I have diligently tried to find a second reviewer for the paper, without success. To avoid further delays, I considered it to be in the author's best interest to receive now the feedback from reviewer 1 and myself. This way, they can already proceed a thorough review of the writing and overall presentation, and improve the manuscript based on the specific feedback given at this point. This will improve the paper for a second round of review, and hopefully gain time in the process.

Detailed comments from reviewer 1 and myself are pasted further below.

We look forward to receiving your revised manuscript.

Kind regards,

Fernanda C. Dórea

Academic Editor

PLOS ONE

Journal Requirements:

Additional Editor Comments:

Please review the writing. The English language, but also the general writing clarity needs to be improved.

In the abstract, for example,

Line 20: remove "system" (or it should be "a system", with "a" added at the beginning of the sentence)

Line 25: yearly?

Line 26: proportion of attendance of what? Some more is explained later in the same paragraph, so maybe just review the paragraph to clarity. Also, attendance "due to" or "attributed to", not "for acute"

Line 34: W/S not explained before what it is.

Abstract and Introduction need rewriting to improve style and clarity.

Methods:

Virological data: you don't need a session for something that was not included. You can just mention at the end of the previous paragraph that this was not considered viable to include.

Other methodological considerations: please see Reviewer 1 comments.

The figures attached had very poor quality, hindering review. The authors mention a lot the 2 waves characteristic of the temporal trend for ARI, I think this should be clearly shown with a plot that has data for all years in the same plot. Maybe the raw data can be presented as a time-series in Figure 1. Preferentially as a bar chart of all consultations (denominator), with a line superimposed showing the proportion attributed to ARI.

Reviewers' comments:

Reviewer's Responses to Questions

**Comments to the Author**

1. Is the manuscript technically sound, and do the data support the conclusions?

Reviewer #1: Yes

2. Has the statistical analysis been performed appropriately and rigorously? 

Reviewer #1: I Don't Know

3. Have the authors made all data underlying the findings in their manuscript fully available?

Reviewer #1: Yes

4. Is the manuscript presented in an intelligible fashion and written in standard English?

Reviewer #1: Yes

5. Review Comments to the Author

Reviewer #1: This is an interesting paper in which the authors model the ARI data from Mauritius with the two methods recommended by WHO for monitoring influenza in the framework of the PISA strategy. This is one of the first studies using MEM two waves method with data from a tropical country.

The paper presents very important findings to help other countries to implement this kind of indicators in their ARI/ILI surveillance systems. However, some questions should be addressed and explained for a better understanding of this work.

Methods.

Line 117 As the data are percentages it is correct the use of the point interval of the arithmetic mean for MEM Intensity thresholds. But you should explain why the Epidemic/seasonal threshold is calculated through the median and Nyblon CI when the arithmetic mean is used (and recommended) in references 14 & 15. It is important also to mention the slope parameter used in the MEM model, and if this parameter was optimized of it has been selected manually.

Line 154. The title of the subsection ‘Sensitivity analysis without the year 2017’ can be confuse. Actually, it is the comparison of the model indicators with and without the year 2017. Maybe ‘Comparison analysis…….’ Anyway, in my opinion PropARI in 2017 are no exceptional: peak around 0.3 (0,2 in 2014), and the authors could consider to present only the model with all the seasons. Or, at least, discuss this point in the discussion section. In this way, the intensity thresholds will reflect better the ARI pattern in the country (the epidemic thresholds and the cross-validation indicators would not change, as the analysis confirm).

Line 176. Statistical analysis. Do you assume that ‘normality’ exists in the cross validation indicators to use the t-test? Please discuss it.

Results

Line 201. Maybe a mistake: The epidemic period for the 2WS curve for MEM (23 weeks) was a week shorter than the associating curve (22 weeks) of WHO ACM.

I would suggest to combine table 2 and table 3 to better describe the text from line 264.

Line 325. It is not clear for me this sentence, since the difference of seasonal thresholds is 0.: Therefore, it is nearly inevitable to observe overlapped confidence intervals of the 2WS curves for WHO ACM and WHO ACM w/o 2017, regardless of the p-value.

Could you explain it a little more?

6. PLOS authors have the option to publish the peer review history of their article (what does this mean?). If published, this will include your full peer review and any attached files.

Reviewer #1: No

---

## [Author Response · Author response to Decision Letter 0]

27 Apr 2021

Fernanda C. Dórea

Academic Editor

PLOS ONE

24th April 2021

Dear Dr. Dórea

Thank you for considering our manuscript for publication and very insightful reviews that we have addressed as shown below. We understand that it has been very hard to find reviewers and we would like to express our gratitude for taking the time to appraise our manuscript.

Kind regards

Mohabeer Teeluck

Epidemiologist/Senior Epidemiologist

Ministry of Health and Wellness, Mauritius

Response to Reviewers

Methods

Question: Line 117 As the data are percentages it is correct the use of the point interval of the arithmetic mean for MEM Intensity thresholds. But you should explain why the Epidemic/seasonal threshold is calculated through the median and Nyblon CI when the arithmetic mean is used (and recommended) in references 14 & 15. It is important also to mention the slope parameter used in the MEM model, and if this parameter was optimized of it has been selected manually.

Answer: Thank you for your review. In reference 14, Vega et al stated the following in the Discussion that “In this work, we chose the parameters which, based on the authors experience, would give the most reliable results. However, these parameters may be adjusted by countries: more/less pre‐epidemic points, median and bootstrap confidence intervals, arithmetic or geometric means, etc. The variations in these parameters do not question the underlying structure of the model and allow the method to be adapted by different surveillance systems.” 

We chose the median and Nyblom CI as we felt it would be more appropriate for comparison of the MEM with the WHO Average Curve method (which calculated the epidemic thresholds using the median formula on Excel spreadsheet) and the fact that we had flexibility to choose this parameter based on the fact that the sample size was greater than 5 (This link https://rdrr.io/cran/mem/man/memmodel.html, the following was stated in the section details at line 13 “Option 3 uses the Hettmansperger and Sheather (1986) and Nyblom (1992) method, when there is enough sample size.” ). 

I would also emphasized that as stated above, that the choice of the parameters can be adapted to our surveillance system and there was no specific recommendation based on the manual for MEM (The MEM Web Application: technical manual Reference 21). 

Question: Line 154. The title of the subsection ‘Sensitivity analysis without the year 2017’ can be confuse. Actually, it is the comparison of the model indicators with and without the year 2017. Maybe ‘Comparison analysis…….’ Anyway, in my opinion PropARI in 2017 are no exceptional: peak around 0.3 (0,2 in 2014), and the authors could consider to present only the model with all the seasons. Or, at least, discuss this point in the discussion section. In this way, the intensity thresholds will reflect better the ARI pattern in the country (the epidemic thresholds and the cross-validation indicators would not change, as the analysis confirm).

Answer: Thank you for your review. We can understand the editor’s concern that the wording of ‘sensitivity analysis’ is confusing. Thus, we have agreed to replace ‘sensitivity analysis’ with ‘comparative analysis’ here as well as in other parts of the manuscript. 

With regards to why we decided that that a peak of 0.3 was exceptional, we even analysed ARI data going back to 2007, and the only time we did find a peak above 0.3 was 0.36 during week 34 in 2009 (N.B: we were in the middle of the influenza H1N1 pandemic of 2009). Therefore, we decided that based on the fact that this event prompted public health actions to be taken by authorities in the country, we compare both models with 2017 and without as a precautionary measure. Therefore, we would appreciate if we could go ahead with the content as it is but as the editor proposed to replace the sensitivity analysis with comparative analysis.

Question: Line 176. Statistical analysis. Do you assume that ‘normality’ exists in the cross validation indicators to use the t-test? Please discuss it.

Answer: Thank you for your insightful review, and based on the fact that the indicators did not follow normality, we have decided to use a non-parametric testing and therefore for this study we chose the Wilcoxon-rank test for all indicators. 

Results

Question: Line 201. Maybe a mistake: The epidemic period for the 2WS curve for MEM (23 weeks) was a week shorter than the associating curve (22 weeks) of WHO ACM.

Answer: Thank you. Corrected. It should have been “The epidemic period for the 2WS curve for MEM (23 weeks) was a week shorter than the associating curve (24 weeks) of WHO ACM.

Question: I would suggest to combine table 2 and table 3 to better describe the text from line 264.

Answer: Table 2 and Table 3 have been merged as recommended.

Question: Line 325. It is not clear for me this sentence, since the difference of seasonal thresholds is 0.: Therefore, it is nearly inevitable to observe overlapped confidence intervals of the 2WS curves for WHO ACM and WHO ACM w/o 2017, regardless of the p-value.

Could you explain it a little more?

Answer: We agree with the assessment of the reviewer that the sentence was not clear and confusing. Therefore we changed the sentence to the following ‘‘Therefore, it was statistically not feasible to verify any significant differences between the 2WS curves for WHO ACM and WHO ACM w/o 2017.’’

Additional Editor Comments:

Line 20: remove "system" (or it should be "a system", with "a" added at the beginning of the sentence)

Corrected as requested

Line 25: yearly?

I have removed the word “weekly” and it’s not yearly. Basically, the proportion of attendance for acute respiratory infections was selected as the transmissibility indicator in this study, to describe seasonality.

Line 26: proportion of attendance of what? Some more is explained later in the same paragraph, so maybe just review the paragraph to clarity. Also, attendance "due to" or "attributed to", not "for acute"

Corrected as requested.

Line 34: W/S not explained before what it is.

Corrected as requested.

Methods:

Virological data: you don't need a session for something that was not included. You can just mention at the end of the previous paragraph that this was not considered viable to include.

Corrected as requested.

Question: The figures attached had very poor quality, hindering review. The authors mention a lot the 2 waves characteristic of the temporal trend for ARI, I think this should be clearly shown with a plot that has data for all years in the same plot. Maybe the raw data can be presented as a time-series in Figure 1. Preferentially as a bar chart of all consultations (denominator), with a line superimposed showing the proportion attributed to ARI.

Answer: New figures uploaded as time-series for figure 3 and figure 4. 

Please review the writing. The English language, but also the general writing clarity needs to be improved.

Abstract and Introduction need rewriting to improve style and clarity.

Answer: We went through the manuscript again and made necessary corrections and did a thorough a grammar check.

---

## [Decision Letter · Decision Letter 1]

21 May 2021

Assessing the appropriateness of the Moving Epidemic Method and WHO Average Curve Method, for the syndromic surveillance of acute respiratory infection in Mauritius

PONE-D-20-28117R1

Dear Dr. Teeluck,

We’re pleased to inform you that your manuscript has been judged scientifically suitable for publication and will be formally accepted for publication once it meets all outstanding technical requirements.

Kind regards,

Fernanda C. Dórea

Academic Editor

PLOS ONE

Additional Editor Comments (optional):

Reviewers' comments:

Reviewer's Responses to Questions

**Comments to the Author**

1. If the authors have adequately addressed your comments raised in a previous round of review and you feel that this manuscript is now acceptable for publication, you may indicate that here to bypass the “Comments to the Author” section, enter your conflict of interest statement in the “Confidential to Editor” section, and submit your "Accept" recommendation.

Reviewer #1: (No Response)

2. Is the manuscript technically sound, and do the data support the conclusions?

Reviewer #1: (No Response)

3. Has the statistical analysis been performed appropriately and rigorously? 

Reviewer #1: (No Response)

4. Have the authors made all data underlying the findings in their manuscript fully available?

Reviewer #1: (No Response)

5. Is the manuscript presented in an intelligible fashion and written in standard English?

Reviewer #1: (No Response)

6. Review Comments to the Author

Reviewer #1: (No Response)

7. PLOS authors have the option to publish the peer review history of their article (what does this mean?). If published, this will include your full peer review and any attached files.

Reviewer #1: **Yes: **Tomás Vega Alonso

---

## [Editor Report · Acceptance letter]

25 May 2021

PONE-D-20-28117R1 

Assessing the appropriateness of the Moving Epidemic Method and WHO Average Curve Method for the syndromic surveillance of acute respiratory infection in Mauritius 

Dear Dr. Teeluck:

I'm pleased to inform you that your manuscript has been deemed suitable for publication in PLOS ONE. Congratulations! Your manuscript is now with our production department. 

Kind regards, 

on behalf of

Dr. Fernanda C. Dórea 

Academic Editor

PLOS ONE